# Interspecific hierarchies from aggressiveness and body size among the invasive alien hornet, *Vespa velutina nigrithorax*, and five native hornets in South Korea

Ohseok Kwon[1], Moon Bo Choi[1,2]*

1 School of Applied Biosciences, College of Agriculture and Life Sciences, Kyungpook National University, Daegu, Republic of Korea, 2 Institute of Agricultural Science and Technology, Kyungpook National University, Daegu, Republic of Korea

* kosinchoi@hanmail.net

**Data Availability Statement:** All relevant data are within the paper and its Supporting Information files.

## Abstract

The range of the invasive alien hornet, *Vespa velutina nigrithorax*, has been expanding since its introduction to Korea in 2003. Here, we compare the aggressive behaviors and body size of *V. velutina nigrithorax* with five native hornet species to identify the interspecific hierarchies that influence the rate of spread of this species. Aggressive behaviors were classified into 11 categories, and each interaction was scored as a win, loss, or tie. We found that *V. velutina* was superior to *V. simillima* in fights that *V. velutina* won and showed a high incidence of threatening behavior. *V. mandarinia* outperformed *V. velutina* in fights that *V. mandarinia* won and grappling behavior was common. *V. analis* was superior to *V. velutina* in fights that *V. analis* won and showed a high degree of threatening behavior. *V. crabro* was superior to *V. velutina* in fights that *V. crabro* won and showed a high rate of threatening behavior. *V. dybowskii* was superior to *V. velutina* in fights that *V. dybowskii* won and showed a high incidence of threatening and grappling behaviors. The body size of *V. velutina* was greater than *V. simillima* (although not statistically significant) and smaller than all other *Vespa* species. Therefore, according to this study, the low interspecific hierarchies of *V. velutina* seem to be a major cause of the slower spread in Korea than in Europe. However, over time, its density has gradually increased within the forest, where it seems to be overcoming its disadvantages and expanding its range, possibly because the large colonies and good flying abilities of this species help it secure food.

## Introduction

In recent years, invasive alien species (IAS) have spread widely due to rapid climate change and global trade, resulting in a global biodiversity reduction, and economic and ecological impacts [1–3]. Indeed, IAS contribute to millions of dollars in economic losses per year [4], with negative impacts from insects and arthropods costing 70 billion US dollars annually [5]. In South Korea, the social and agricultural impacts are also gradually increasing due to

**Funding:** The author(s) received no specific funding for this work.

**Competing interests:** The authors have declared that no competing interests exist.

invasions by black widow spiders (*Latrodectus hesperus*), spotted lanternfly (*Lycorma delicatula*), frosted moth-bug (*Metcalfa pruinosa*), leaf-footed bug (*Leptoglossus gonagra*) and black planthopper (*Ricania speculum*). In particular, social insects such as yellow-legged hornets (*Vespa velutina*), red imported fire ants (*Solenopsis invicta*), and argentine ant (*Linepithema humile*), which are particularly damaging due to the large number of individuals and their toxicities, are recent introductions to Korea and pose a significant social threat [6–13].

Many IAS spread rapidly and broadly after the successful invasion of new environments through resource and habitat competition with native species [14–17]. In particular, there are many cases of social insect invasions worldwide, such as in ants, wasps, and bees, and if the invasion is successful, it can have serious ecological and economic impacts, with population sizes ranging from hundreds to tens of thousands of individuals [18].

*Vespa velutina nigrithorax*, originating in southern China, has spread throughout South Korea since its first invasion in 2003, where it was introduced through trade ships [8, 19–21]. After invading Tsushima Island in Japan in 2012, *V. velutina* invaded Kyushu in mainland Japan in 2015 [22–24]. In Europe, this species spread to France, Spain, Portugal, Italy, Germany, Belgium, Switzerland and the UK by 2016 after the first invasion in France in 2004 [25–27]. *V. velutina* has a severe economic impact on beekeepers by foraging large quantities of honeybees in apiaries, removing approximately 30% of honeybee colonies [28].

In addition, *V. velutina* is a poisonous insect that has a public health impact. In Korea, there are more than 100,000 cases of removal of social wasps' nests per year, and *V. velutina*'s nest removal rate is the highest among *Vespa* species. The average number of injuries caused by social wasps is about 15,000, and there have been about 10 deaths. In particular, due to the high density in urban areas, the damage caused by them is likely to be high [19, 20, 29]. In fact, two deaths have occurred in France since the invasion of *V. velutina*, and two deaths have been reported in Korea. At present, the impacts of *V. velutina* may not be noticeable [30], but the actual impacts are expected to be more because the extent of the impacts that this alien species causes may not be fully appreciated [28, 31–33].

This species also causes ecosystem disturbance through competition and interference with other *Vespa* species in nature [34–36]. Therefore, *V. velutina* shows the comprehensive impact of IAS. In Korea, *V. velutina* was designated as an Ecological Disturbance Organism (Ministry of Environment Notice 2019–185) on 26 July 2019 under the provisions of Article 23 of the Act on the Conservation and Use of Biodiversity (http://www.law.go.kr/LSW/admRulLsInfoP.do?chrClsCd=&admRulSeq=2100000180728, access date November 14, 2019).

Obtaining adequate nutrition is the most important factor for the primary survival and range expansion of *V. velutina* colonies. In general, social wasp prey differs between adult and larval stages. Larvae require protein for growth, and obtain this by being fed by adults, which hunt. Adults, on the contrary, consume carbohydrates in order to obtain energy due to their higher levels of activity. Therefore, in nature, adults eat oak sap, flower honey, and nectar [37]. In particular, oak sap comes from various butterflies and flies, as well as medium and large beetles such as dynastid beetles, stag beetles and weevils [38–40]. Some insects eat sap with other insects around the sap, but hornets and large beetles compete for limited sap resources [37, 38]. This competition also occurs among several *Vespa* species, where *V. mandarinia* is predominantly high hierarchy in regard to Japanese oak sap, followed by *V. crabro*, *V. analis*, and *V. simillima* [37]. The results of competition between these species, therefore, are very helpful in identifying the species' ecological niche in the ecosystem.

Several factors determine the rate of spread of an invasive species. In addition to anthropogenic controls [1, 41, 42] and natural enemies [43, 44], the ecological hierarchy obtained through competition among similar species makes an important contribution to the likelihood and speed of range extension.

Therefore, in this study, we analyzed the interspecific hierarchies of *V. velutina* among Korean *Vespa* species by measuring aggressive behavior to secure food sources among native hornet species and *V. velutina*, in order to understand competitive ability, which is the main factor determining the successful spread of *V. velutina*. In addition, we analyzed the correlation between fighting ability and size by measuring the body size of each species.

## Materials and methods

### Study species and experiment sites

We planned to test for aggressive behaviors between *V. velutina* and nine native Korean *Vespa* species, but because of their different distributions, it was impossible to observe the behavior of all nine species in one place. In particular, *Vespa simillima xanthoptera*, *Vespa binghami*, and *Vespa crabro crabroniformis* have very limited distributions, making them difficult to compare [45]. Therefore, experimental sites at which many *Vespa* species, including *V. velutina*, are present were selected by referring to various studies such as those of Choi et al. [45] and Choi and Kwon [46]. As a result, we conducted the experiment in the Piagol Valley of Jirisan National Park, where seven *Vespa* species (*V. velutina*, *V. simillima*, *V. crabro*, *V. dybowskii*, *V. mandarinia*, *V. analis*, and *V. ducalis*) occur.

First, we randomly chose three experimental sites in Piagol Valley (Site A: N35 ˚ 13'37.96 "E127 ˚ 35'48.95", 174m; Site B: N35 ˚ 15'17.53 "E127 ˚ 35'58.24", 417m; Site C: N35 ˚ 15'42.43 "E127 ˚ 35'3.57", 398m, Since this experiment was conducted on a private site outside the boundaries of a Jirisan National park, it was conducted with the personal permission of the landlord. Therefore, there is no documented permit). *Vespa* species were captured using hornet traps in the apiaries near each site, and identified. As a result, four species (*V. velutina*, *V. simillima*, *V. mandarinia*, and *V. crabro*) were captured in the apiary near site A, six species (*V. velutina*, *V. simillima*, *V. dybowskii*, *V. mandarinia*, *V. analis*, and *V. crabro*) were captured in the apiary near site B, and five species were captured in the apiary near site C (*V. velutina*, *V. simillima*, *V. mandarinia*, *V. analis*, and *V. crabro*). Therefore, site B was selected as the experimental site, as it had the highest species diversity. *V. ducalis* was also seen flying near site B but did not appear in traps or the test site. Therefore, the behavioral experiment was conducted between *V. velutina* and five native hornet species.

### Behavioral observation experiment

The behavioral observation experiment apparatus was set up as follows: a table (1m in height) was installed on a flat plain in a forest where hornets were present, 2–3 sheets of toilet paper were placed on top of the table, and an attractant was poured onto these, following the methods described by Choi et al. [19]. In addition, to increase hornet attraction, a 500-ml nebulizer was filled with attractant, and the liquid was sprayed for approximately 10–20 minutes before observations began. The attractant was composed of 1: 1: 1 brown sugar water, vinegar, and ethanol, which mimics oak sap. It is the most commonly used substance for attracting and capturing Vespinae species in hornet traps in Korea, and it has very little attractiveness bias for a particular species [19, 47].

This experiment was conducted only with workers, excluding gynes and males, to get rid of the differences in attack behaviors due to caste differences. Therefore, we conducted our study over a total of four days from August 12–13 and August 17–18, 2017, before the gynes and males came out in mid-September. The experiment was conducted at 8–10 am and 5–7 pm because workers tend to avoid outdoor activities when daytime temperatures exceed 35 degrees. The experiment was conducted eight times in total.

## Behavioral description, intensity scores, winning percentages, and aggressive behavior trends

As there were no previous behavioral descriptions of the aggressive behavior of hornets, we prepared the first description based on Jang et al. [48] and Goyens et al. [49], by analyzing images from our experiment. Images of hornet behavior were taken by recording hornets with a camcorder (Digital Camcorder V 2000, Sunwoo Tech. Crop. Goyangsi, Korea), placed 50 cm in front of the experiment table.

Aggressive behavioral interactions were classified as wins, losses, and ties. Losses was given 0 points, and ties given 1 point. For wins, threatening behaviors were given 2 points, grappling behaviors were given 3–4 points, and killing was given 5 points. These intensity scores are summarized in Table 1. Winning percentages were calculated, showing the percentage of wins, losses, and ties. In addition, the tendency towards aggressive behavior between the two species was expressed in the radar chart of the number of each behavior. The score for each behavior is derived from Table 1 and Fig 2.

## Morphological measurements

To measure the body size of the six species, we captured them using an insect net following the behavioral observation experiment and stored them in 95% alcohol. Both winners and losers were included in the sampling. If the test subjects were missed, the remaining opponents were excluded from the measurement. Captured samples were dried at room temperature (20˚C—25˚C) for one week in the laboratory and then pinned. The lengths (mm) of the specimens were obtained by measuring only the head width and thorax length through a stereoscopic microscope (Olympus optical, JP SZ61, Olympus Korea, Seoul, Korea), as the hornet's abdomen was elastic and could affect its body size. Body size of the five native species was measured by selecting 30 individuals from the samples collected during the experiment. In addition, *V. velutina* was selected from the individuals who fought the five native species, and a total of 30 individuals were measured.

## Statistical analysis

We performed an independent t-test on the intensity scores of *V. velutina* and the five native hornets to verify the significance of any differences in aggression between species.

**Table 1. Descriptions of hornet aggression behaviors observed and their corresponding scores.**

| Behavior | Description | | | Intensity score | | |
|---|---|---|---|---|---|---|
| | **Win** | **Loss** | **Tie** | **Win** | **Lose** | **Tie** |
| Threatening | Rushing opponent | Falling back or escaping | Confrontation and separation | 2 | 0 | 1 |
| | Lifting antennae and front legs and shaking wings, (Fig 1B) | Falling back or escaping | Confrontation | 2 | 0 | 1 |
| | Opening mandible, (Fig 1F) | Falling back or escaping | Confrontation | 2 | 0 | 1 |
| | Threateningly flying over opponent | Escaping | Disregard or confrontation | 2 | 0 | 1 |
| | Chasing opponent | Escaping | Confrontation and separation | 2 | 0 | 1 |
| Grappling | Pushing or fighting while flying | Falling back or escaping | Confrontation and separation | 3 | 0 | 1 |
| | Banging or pushing with head | Falling back or escaping | Confrontation and separation | 3 | 0 | 1 |
| | Chasing and grabbing, (Fig 1E) | Being thrown off | Confrontation and separation | 3 | 0 | 1 |
| | Forcing down and throwing opponent, (Fig 1D) | Being thrown off | Confrontation and separation | 3 | 0 | 1 |
| | Getting opponent and biting or stinging, (Fig 1A) | Being bitten or stung | Confrontation and separation | 4 | 0 | 1 |
| Killing | Killing opponent, (Fig 1C) | Being killed | Failure to kill opponent or both sides dying | 5 | 0 | 1 |

In addition, the size differences among the six hornets, including *V. velutina* were analyzed using a one-way analysis of variance (ANOVA) with a post hoc Tukey's honest significant difference (HSD) test. All analyses were performed using IBM SPSS Statistics software version 23.0 (IBM, USA).

## Results

### Description of aggressive behavior

Aggressive behaviors between *V. velutina* and native hornets were classified into three categories: threatening, grappling, and killing. The full list of all behaviors within these categories are listed in Table 1.

First, threatening occurred when the two individuals maintained a distance from each other and ate the attractant, and as the distance between them became closer, they threatened their opponents without making direct contact. The following behaviors were considered threatening: an individual moving forward, lifting its antennae and front legs, shaking its wings, or opening its mandible (Fig 1B and 1F). In these cases, the individual was considered to have won when the opponent fell back or ran away. An individual flying above its opponent, or chasing the opponent, were also considered threatening behaviors. In these cases, the individual was considered to have won when the opponent ran away. Interactions were considered a tie when opponents threatened and fell behind each other.

Grappling occurred between two individuals if the distance between them became very narrow or touched. They flew into the air, either fighting or chasing each other and throwing their opponent. Pushing their opponent's head, hitting, throwing, climbing on top of, biting, or attacking their opponent with a stinger were all considered grappling behaviors (Fig 1A, 1D and 1E). In these interactions, the individual who fell back, ran away, fell, or was bitten was

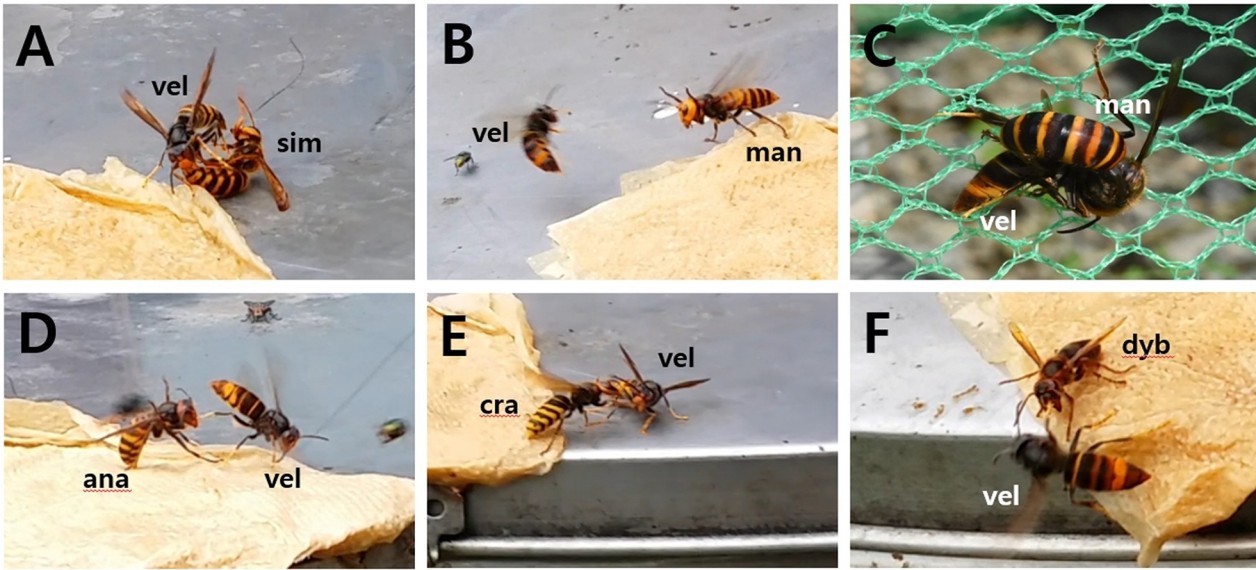

**Fig 1. Aggressive behavior between an invasive alien hornet, *Vespa velutina*, and five native Korean hornet species (*Vespa simillima*, *Vespa mandarinia*, *Vespa analis*, *Vespa crabro*, and *Vespa dybowskii*).** A: biting, *V. velutina* bites *V. simillima* using its mandibles, B: lifting antennae and shaking wings, *V. mandarinia* threatens *V. velutina* by raising its antenna and shaking its wings as *V. velutina* approaches, C: killing, *V. mandarinia* hunts *V. velutina*, D: forcing down and throwing, *V. analis* throws *V. velutina*, E: banging or pushing with head, *V. crabro* pushes *V. velutina* with its head, F: open mandible, *V. dybowskii* threatens *V. velutina* with its mandibles. Species codes: vel: *V. velutina*, sim: *V. simillima*, man: *V. mandarinia*, ana: *V. analis*, cra: *V. crabro*, dyb: *V. dybowskii*.

considered to have lost. If the individuals fall back behind each other while facing each other, this was a tie.

Finally, killing occurred when an overwhelmingly strong individual fought with an opponent, and killed it (Fig 1C). If predation failed or both individuals died, this was a tie.

## Aggressive intensity, winning percentages, and aggressive behavior trends

In *V. velutina* vs. *V. simillima*, *V. velutina* was superior to *V. simillima* in 153 fights ($t_{(304)}$ = 9.89, P < 0.001, Fig 2A), and won 71% of the encounters (Fig 3). *V. velutina* had a high rate of threatening behavior, such as rushing the opponent, and lifting the antennae and front legs and shaking the wings. *V. simillima*, however, rushed the opponent more often than *V. velutina* (Fig 4A).

In *V. velutina* vs. *V. mandarinia*, *V. mandarinia* outperformed *V. velutina* in a total of 104 fights ($t_{(206)}$ = -22.75, P < 0.001, Fig 2B) and won 91% of the encounters (Fig 3). *V. mandarinia* displayed a high rate of grappling behavior, such as banging or pushing with the head (Fig 4B). *V. mandarinia* also preyed upon *V. velutina*, although rarely.

In *V. velutina* vs. *V. analis*, *V. analis* was superior to *V. velutina* in 67 fights ($t_{(132)}$ = -8.38, P < 0.001, Fig 2C) and won 76% of the encounters (Fig 3). Concerning threatening behavior, *V. analis* displayed threat most often by rushing the opponent, and this species' second favorite

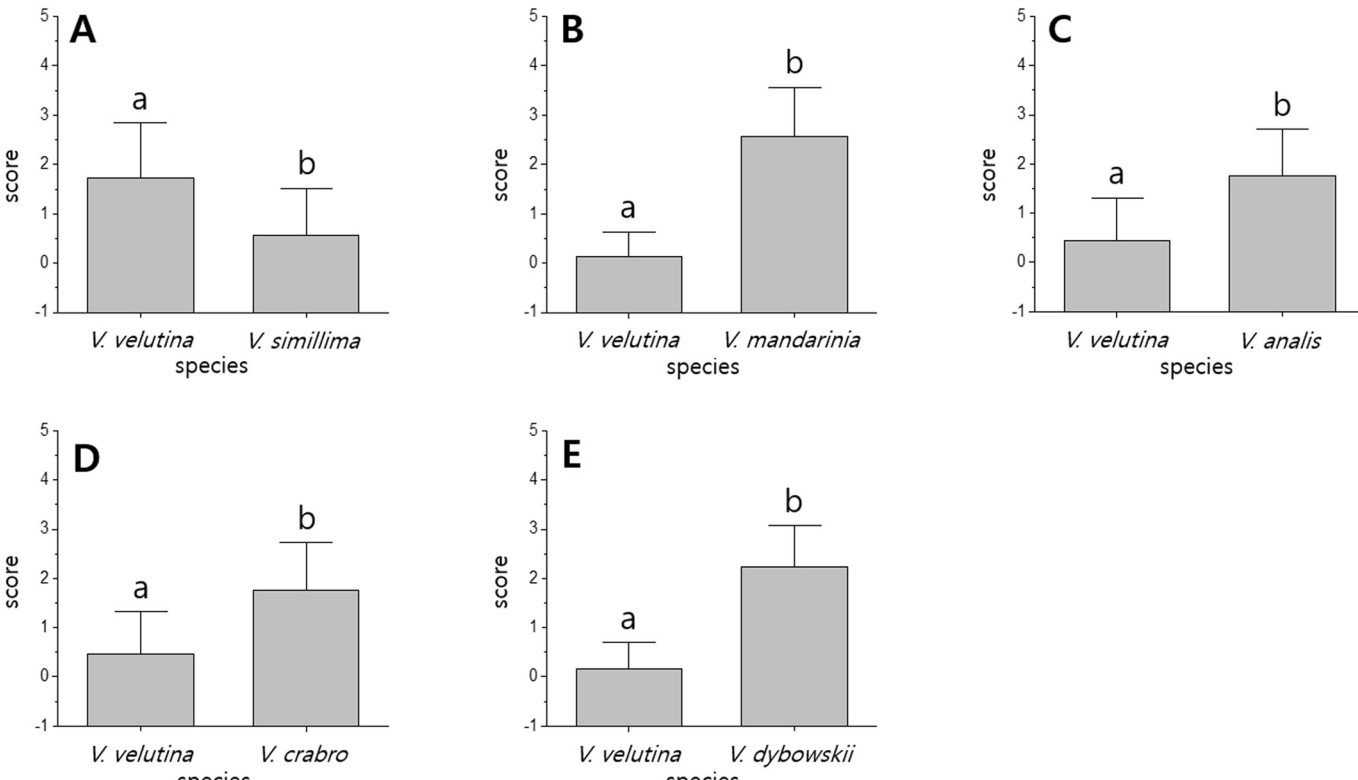

**Fig 2. Aggressiveness scores between an invasive alien hornet, *V. velutina*, and five native Korean hornet species.** Scores are displayed as mean values ± SD. A: *V. velutina* (score 1.73±1.11) vs *Vespa simillima* (score 0.57±0.94), $t_{(304)}$ = 9.89, P < 0.001; B: *V. velutina* (score 0.13±0.48) vs *Vespa mandarinia* (score 2.58 ±0.98), $t_{(206)}$ = -22.75, P < 0.001; C: *V. velutina* (score 0.45±0.86) vs *Vespa analis* (score 1.76±0.96), $t_{(132)}$ = -8.38, P < 0.001; D: *V. velutina* (score 0.47±0.86) vs *Vespa crabro* (score 1.76±0.97), $t_{(184)}$ = -9.62, P < 0.001; E: *V. velutina* (score 0.16±0.54) vs *Vespa dybowskii* (score 2.24±0.83), $t_{(262)}$ = -24.22, P < 0.001. See S1 Table.

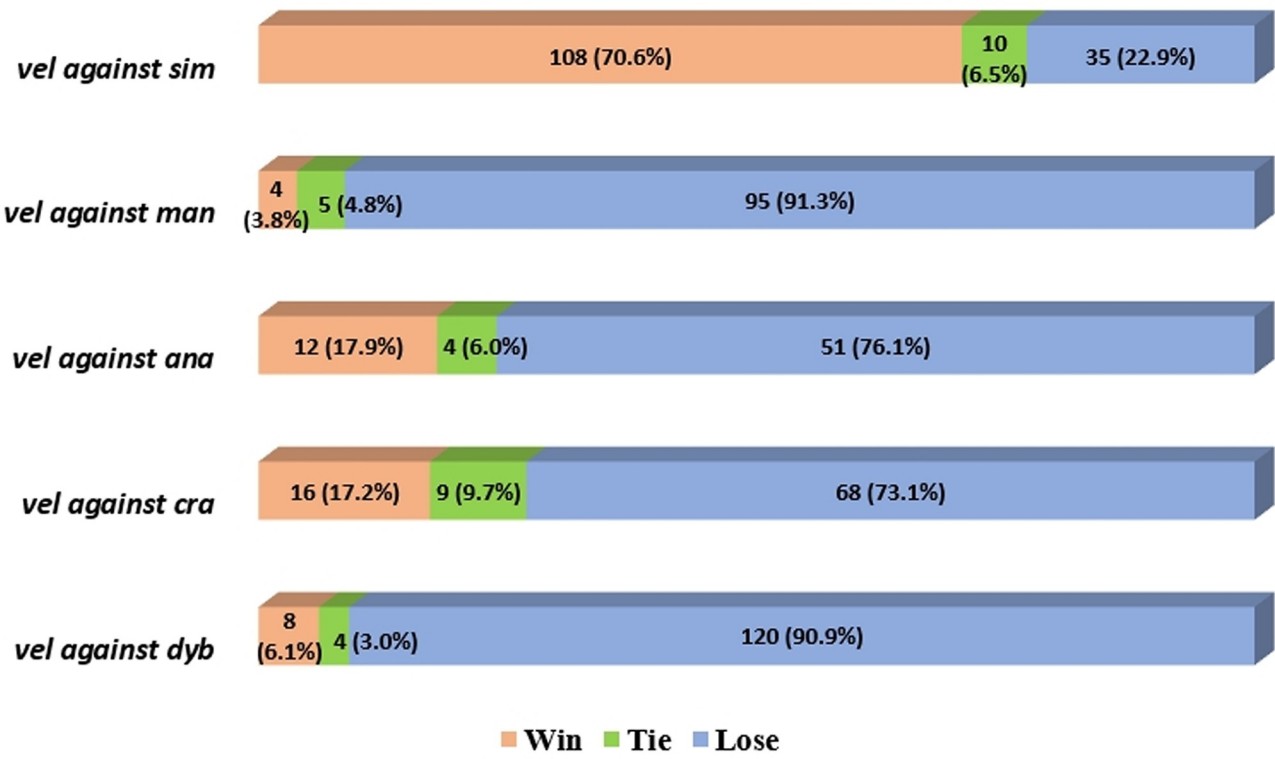

**Fig 3. Winning percentages between an invasive alien hornet, *V. velutina*, and five native Korean hornet species.** All wins and losses are based on *V. velutina*. Species codes: vel: *V. velutina*, sim: *V. simillima*, man: *V. mandarinia*, ana: *V. analis*, cra: *V. crabro*, dyb: *V. dybowskii*.

ploy was to fly threateningly over the adversary. In contrast, *V. velutina* made a weak show of rushing the opponent (Fig 4C).

In contests between *V. velutina* and *V. crabro*, *V. crabro* was superior to *V. velutina* in 93 fights ($t_{(184)}$ = -9.62, P < 0.001, Fig 2D) and won 73% of the encounters (Fig 3). Lifting antennae and front legs and shaking wings were the most common threatening behaviors in *V. crabro*, and banging or pushing with the head were the most common grappling behaviors. In contrast, *V. velutina* displayed minor aggressive behavior such as threateningly flying over the opponent (Fig 4D).

Finally, in encounters between *V. velutina* and *V. dybowskii*, *V. dybowskii* was superior to *V. velutina* in a total of 132 fights ($t_{(262)}$ = -24.22, P < 0.001, Fig 2E) and won 91% of the contests (Fig 3). The most common threatening behavior employed by *V. dybowskii* was chasing the opponent, and the second most common action was rushing the opponent; chasing and grabbing were the most common grappling behaviors for this species. *V. velutina*'s counterattack action was to fly threateningly over the opponent but not vigorously or often (Fig 4E).

## Body size

Numerically, the body size of *V. velutina* was 8.04 ± 0.41 mm, which was slightly larger than *V. simillima* 7.80 ± 0.29 mm. *V. dybowskii* was 8.88 ± 0.49 mm, *V. analis* was 10.14 ± 0.51 mm, *V. crabro* was 9.82 ± 0.4 mm, and *V. mandarinia* was 13.21 ± 0.83 mm. Thus, except for *V. simillima*, all species were larger than *V. velutina*. There were significant differences in body size among *Vespa* species ($F_{(5, 174)}$ = 434.9, P < 0.001), being the differences significant between all the species except for *V. simillima* and *V. velutina*, and *V. analis* and *V. crabro* (Fig 5).

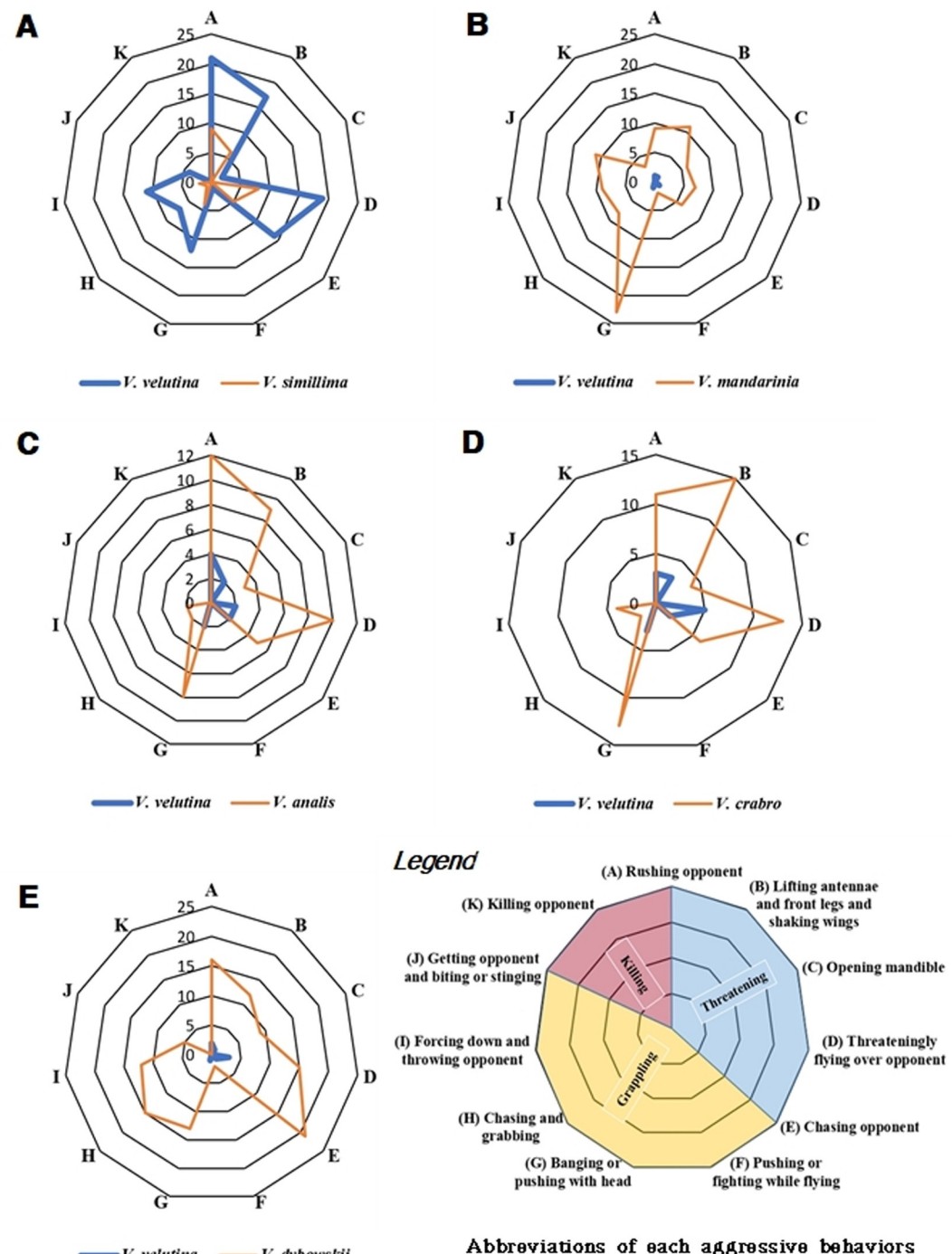

**Fig 4. Trends of aggressive behaviors between an invasive alien hornet, *V. velutina* and five native Korean hornet species.**
The aggressiveness score of behaviors increases clockwise and the score of each behavior is the sum of the intensity score.

## Discussion

### Spread effect of *V. velutina* on interspecific competition in Korea

The invasion of IAS generally results in interspecific competition for resources with similar native species [50]. If the native species are strong, invasive species tend to avoid competition

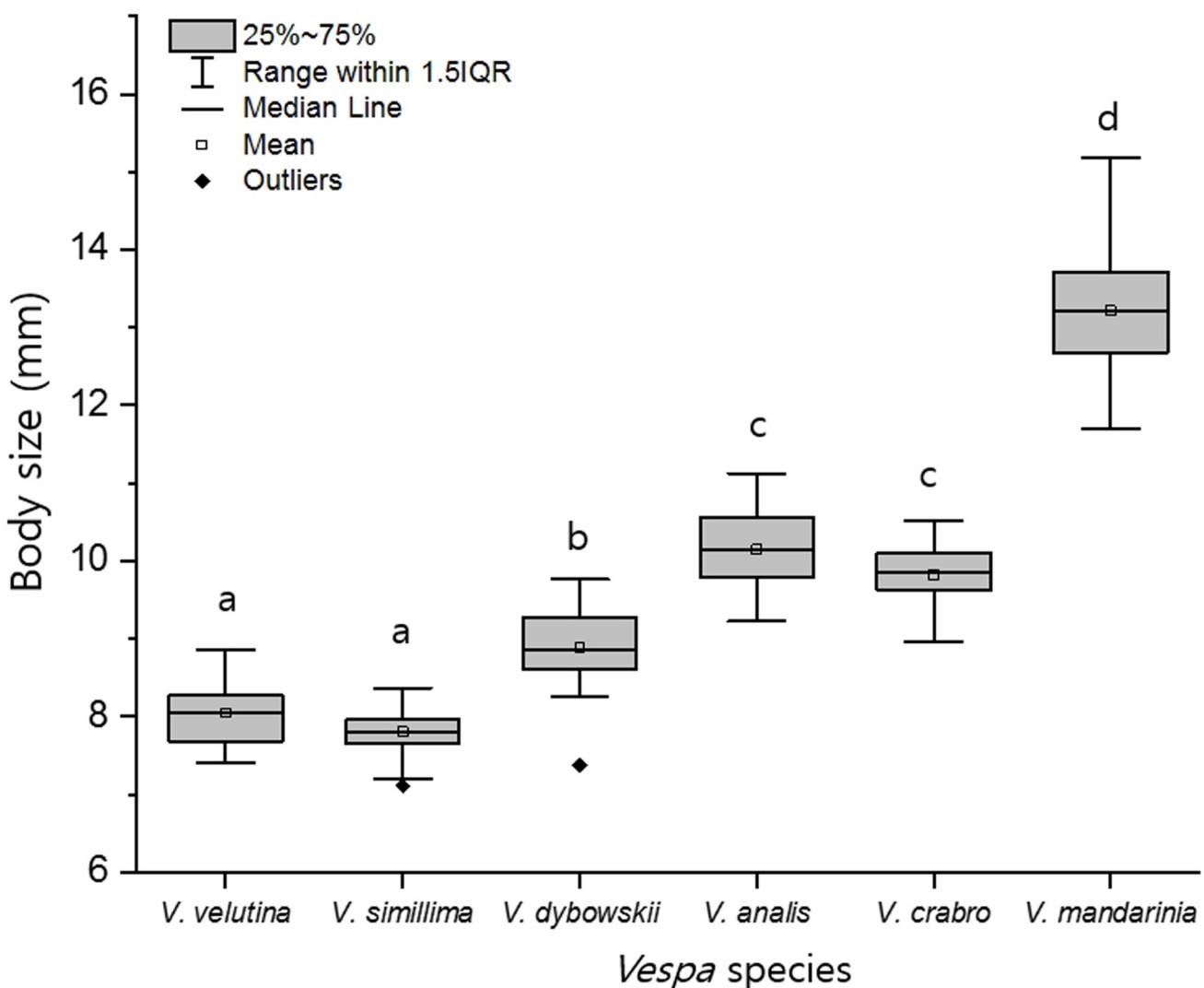

**Fig 5. Body size differences between an invasive alien hornet, *V. velutina* and five native Korean hornet species.** $F_{(5, 174)}$ = 434.9, P < 0.001.

temporally or spatially, but if the native species are weak or similar, exploitation or competition may interfere [29, 51], resulting in interspecific displacement [50].

In Europe, where *V. velutina* was introduced in 2004, there were already two native hornet species, *V. crabro* and *V. orientalis*, so that the two species were in close competition because of their similar ecological niche, due overlaps in food preference, nesting sites, and life history [34, 35]. However, in Korea, where there are nine native hornets, *V. velutina* clearly differ in their ecological niche. *V. mandarinia* is the most aggressive among hornets, with the highest hierarchy, nesting in the ground, and mainly foraging on honeybees and large beetles [37]. *V. crabro* nests in the ground, under trees, and *V. analis* nests under grasses and leaves, hunting small and medium-sized insects such as bees, flies, and moths [37]. *V. dybowskii* nests in tree trunks and often display social parasitism [37]. *V. velutina*, however, nests at the tops of trees, with a lifespan of about one month longer and a higher rate of honeybee foraging [28, 52]. *V. simillima* are phylogenetically closest to *V. velutina* [53] and have the most ecological overlap because of similarities in life history, nesting, and food preference. Therefore, *V. simillima* may be most severely

affected by *V. velutina* [19]. According to the results of this study, *V. velutina* is less aggressive than the other four species except *V. simillima*, indicating that the ecological niche is low. Therefore, *V. velutina* avoids competition from four species with strong ecological niche when nesting, as it nests in a place with low competition. Indeed, after the invasion of *V. velutina* in Busan, Korea, there was a decrease in *V. similima* [19]. Therefore, after *V. velutina*'s invasion of Korea, its distribution and spread seem to be affected by its ecological niche. From Busan, the point of first invasion of *V. velutina*, it spread northwards into cities such as Ulsan, Gimhae, and Changwon, where the road network is directly connected, and the east and south coasts [19, 45, 54]. Since there are nine native hornets within Korean forest habitats, *V. velutina* would have had considerable difficulty invading deep forests and crossing large forests and mountains [54].

This ecological niche is often determined by differences in fighting ability, which is largely influenced by body size. In many animals, large body size is likely to corelate with physical fighting ability, which can lead to intraspecific and interspecific resource competition. This is termed resource-holding potential (RHP) [55, 56]. For example, in various insects, larger individuals in the intraspecific have higher aggressiveness and higher RHP than small ones, indicating that they are competitive [48, 49, 57, 58]. Large invasive species have been shown to spread more easily in nature with competitive advantages at higher RHP than smaller native species [59]. Therefore, in this study, as *V. velutina* is a similar size to or larger than *V. simillima*, but smaller in size than the other four native hornets, its RHP seems to be relatively low, as shown in the aggressiveness results of this study.

Therefore, *V. velutina* appears to avoid direct competition with native hornets, inhabiting urban centers with relatively low-density hornet populations, or on mountain edges adjacent to cities. According to Choi et al. [19], in Busan, the occurrence rate of *V. velutina* was overwhelmingly higher in cities than in forests.

Meanwhile, the low interspecific hierarchy of *V. velutina* may have influenced their initial spread. In general, the initial incubation period is determined according to the presence or absence of competing species after the invasion of alien species, and after a certain period of time, when the alien species have a firm ecological niche, the population expands and spreads rapidly [60]. Thus, in Europe where the competition was *V. crabro* at the beginning of the invasion [34, 35], the annual spread rate was 60–80 km/year [61, 62]. In Korea, however, the rate of spread is slower than that of Europe at 10–20 km/year, because 63% of the land is forested, and nine *Vespa* species already inhabit forests [19, 63, 64]. The forest landscape of Korea and presence of many competing hornet species had the effect of prolonging the incubation period. Thus, for *V. velutina*, one of the main reasons why it could survive without natural extinction, despite its low ecological niche, seems to be its choice of habitat in cities with low competition among hornets during incubation period.

*V. velutina* has ended its initial incubation period, and the rate of spread has increased rapidly from 2008–2010 [64], culminating in its distribution throughout South Korea as of 2018 [20]. Of course, this spread may have been affected by human factors, such as human activity or transport [61], but competition among hornets seems to be the main reason.

However, despite its low aggression and small size, *V. velutina* population density tended to increase not only in urban areas but also in forested areas over time, and this trend is present in most parts of southern-central Korea [46].

Many invasive alien social insects, such as ants, hornets, yellowjackets, and paper wasps, have large colonies, high polygyny rates, or invasive generalist insect predators, which often lead to dominant distribution and species displacement with successful invasion. This is because alien species with larger colonies than native species are much easier to feed [16, 17, 65–69]. Although *V. velutina* has a lower ecological niche than native hornets, the colonies are two to four times larger [19, 70], which is an overwhelming number of individuals that can

obtain food resources in nature. In addition, *V. velutina* has a good flight ability compared to other hornets [71, 72] and hunts for food quickly and safely through hawking [46, 73], making it very efficient. Altering the concentration and composition of alarm pheromones in areas it has invaded also creates an effective defense strategy against potential predators [74, 75]. Therefore, in the early stages of the invasion, it spread to urban areas, competing with native hornets, but due to its large colonies and high foraging ability, they seemed to overcome their disadvantages and gradually increase their abundance even in the forest.

### Interspecific hierarchies of Korean *Vespa* species

In this study, the interspecific hierarchies of Korean *Vespa* species was determined in part through the aggression behavior of six *Vespa* species, but the remaining four Korean species were not tested [45]. However, of these four species, *V. crabro crabroniformis* is a subspecies of *V. crabro* (the *V. crabro* used in this experiment is *V. crabro flavofasciata*), and *V. simillima xanthoptera* is a subspecies of *V. simillima* (the *V. simillima* used in this experiment was *V. simillima simillima*). These are very similar to the two species we observed, so we expect there to be little difference in ecological characteristics between them. In particular, *V. simillima xanthoptera* only inhabits Jeju Island in Korea, and has no geographical overlap with the mainland species. *V. binghami* was first reported in Korea in 2011 [76], and there is a lack of ecological information on this species, which requires further study. Finally, *V. ducalis* was not addressed in this study, but Yoshimoto and Nishida [39] showed it was less aggressive than *V. crabro* and *V. analis*, and Matsuura [77] showed it was less aggressive than *V. simillima*. It is therefore likely to be the least aggressive among the *Vespa* species.

As a result, according to Matsuura and Yamane [37], the order of hierarchies *Vespa* species in Japan is *V. mandarinia* > *V. crabro* > *V. analis* > *V. simillima*.

However, in this study, our results suggested the order for Korean *Vespa* species should be *V. mandarinia* > *V. dybowskii* > *V. analis* > *V. crabro* > *V. velutina* > *V. simillima* (Fig 6).

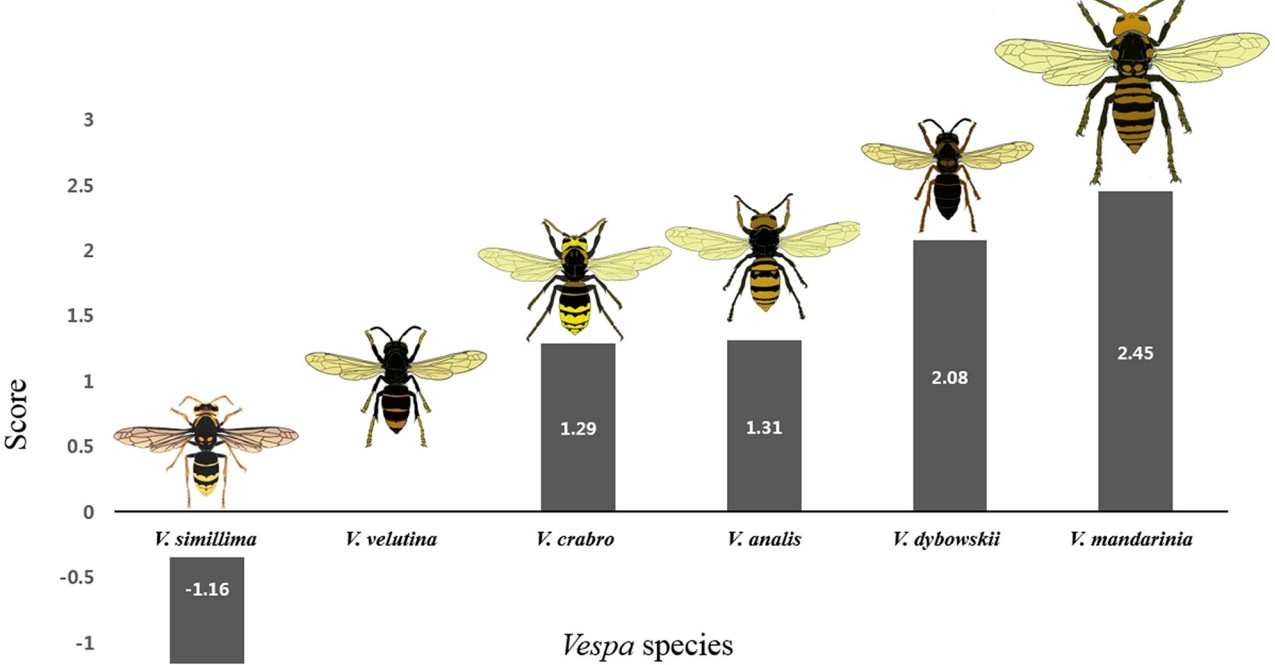

**Fig 6. Interspecific hierarchies of an invasive alien hornet *V. velutina* and five native Korean hornet species inferred by aggressiveness scores.**

Although the order of hierarchies *V. analis* and *V. crabro* changed in comparison with the order in Matsuura and Yamane [37], the sequence is similar. If *V. ducalis* were to be added here, we would expect the order to be *V. mandarinia* > *V. dybowskii* > *V. analis* > *V. crabro flavofasciata* (or *V. v. crabroniformis*) > *V. velutina* > *V. simillima* (or *V. s. xanthoptera*) > *V. ducalis*.

## Supporting information

**S1 Table. Distribution of aggressiveness scores between an invasive alien hornet, *V. velutina*, and five native Korean hornet species.**
(DOCX)

## Acknowledgments

This study is a part of MB Choi's Ph. D study.

## Author Contributions

**Conceptualization:** Moon Bo Choi.

**Formal analysis:** Ohseok Kwon, Moon Bo Choi.

**Investigation:** Moon Bo Choi.

**Methodology:** Ohseok Kwon, Moon Bo Choi.

**Resources:** Moon Bo Choi.

**Supervision:** Moon Bo Choi.

**Validation:** Ohseok Kwon.

**Visualization:** Moon Bo Choi.

**Writing – original draft:** Ohseok Kwon, Moon Bo Choi.

**Writing – review & editing:** Ohseok Kwon, Moon Bo Choi.

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
