## [Decision Letter · Decision Letter 0]

13 Feb 2020

PONE-D-19-34041

Interspecific hierarchies from aggressiveness and body size among the invasive alien hornet, Vespa velutina nigrithorax, and five native hornets in South Korea

PLOS ONE

Dear Dr. Choi,

Thank you for submitting your manuscript to PLOS ONE. After careful consideration, we feel that it has merit but does not fully meet PLOS ONE’s publication criteria as it currently stands. Therefore, we invite you to submit a revised version of the manuscript that addresses the points raised during the review process.

We would appreciate receiving your revised manuscript by Mar 29 2020 11:59PM. To enhance the reproducibility of your results, we recommend that if applicable you deposit your laboratory protocols in protocols.io, where a protocol can be assigned its own identifier (DOI) such that it can be cited independently in the future. For instructions see: http://journals.plos.org/plosone/s/submission-guidelines#loc-laboratory-protocols

We look forward to receiving your revised manuscript.

Kind regards,

Amparo Lázaro, PhD

Academic Editor

PLOS ONE

Journal Requirements:

Reviewers' comments:

Reviewer's Responses to Questions

**Comments to the Author**

1. Is the manuscript technically sound, and do the data support the conclusions?

Reviewer #1: Yes

Reviewer #2: Partly

2. Has the statistical analysis been performed appropriately and rigorously? 

Reviewer #1: Yes

Reviewer #2: Yes

3. Have the authors made all data underlying the findings in their manuscript fully available?

Reviewer #1: Yes

Reviewer #2: Yes

4. Is the manuscript presented in an intelligible fashion and written in standard English?

Reviewer #1: Yes

Reviewer #2: No

5. Review Comments to the Author

Reviewer #1: This is an interesting study which analyzes how aggressiveness of different native hornets may influence the invasiveness of Vespa velutina, a very invasive species in several countries since 2003-2004, and which now causes severe damage to honeybees in Europe.

The study is limited to the South Korean area but it has merit of novelty by scoring interspecific hierarchies between 6 differents species. this local study may help analyzing similar patterns in other countries.

Manuscript is well written, data and literature correctly analyzed.

Several details may help improving the manuscript.

Intro first § focus on several invasive species in Korea. What about the invasive status of the other hornets. e.g. crabro is known as also invasive in several countries. This § could be extended somehow.

About the health impact. This impact on health is to my opinion not completely founded. A study in france (De haro et al.) published that cases of hymenopteran evenomations did not increase in france after Vv invasion, and only two cases of death were clearly identified (venom identified by the police scientific services). The main impact on health is of course due to shock in case of multiple sting when attacking a nest or allergic reaction (same with honey bees or yellow jackets).

What coul also be mentionned is the frenzy to citizens when nests are in private gardens or parks.

This § could be improved.

details concerning mandatory in Korea should not appear in the full text since it concerns only South Korea. Could this be in additional data ?

I do not fully agree with one of the last intro sentence : 'in particular, the possibility and speed of IAS spread... are detremined by the hierarchy...' Not only, biocontrol agents localy presents and natural ennemies may also wipe out the IAS during the first steps.

This should be rephrased (please quote example of IAS control by natural ennemies (including disease). This values also for the begining of discussion.

The role of venom gland (as so caleld alarm pheromone in two recent papers should also be documented and discussed somewhere.

Reviewer #2: The paper investigates the interspecific aggressiveness and body size among Vespa velutina, an invasive alien hornet, and five native hornets in South Korea.

Although it is known that some hornet species are more aggressive than others and the body size can matter for this, the interspecific aggressive relationship among congeneric and sympatric species is not deeply investigated so far.

The study falls in the recent line of studies investigating the invasiveness potential of Vespa velutina. To assess this topic, the Authors carried out tests of aggressive behaviour in the field and measured the body size of each species. I find the topic and the results interesting, but I have some suggestions to improve clarity and some concerns about methods that should be taken into account before publication.

My comments and suggestions are listed below (unfortunately the Ms does not reported lines number to help in the revision process); the authors should address such comments before the manuscript is suitable for publication.

ABSTRACT

The abstract needs to be rewritten by summarizing in a few lines the results by cutting all numbers and percentage of fights.

METHODS

Behavioral observation experiment

…..in a forest where hornets were active: what do you mean for “active”? I think it means “present”.”

….to spray the attractant for approximately 10-20 minutes : Please, give the quantity of the attractant sprayed. I believe that it should be better reported the composition here rather than the reference that reported it.

It is unclear if the Authors sprayed the attractant every 10-20 min or they spayed the attractant in the air for 10-20 min before beginning the observations to increase the hornets attraction?

Moreover, I have some concern about the attractant used. It seems to be a generic one but the different species could be differently specialized as reported by Matsura (1991). This could bias the experiment as one species could be more aggressive than others if it must defence a resource more . Have the Authors some data about the attractiveness level for each species?

….. because new queens or males were confused with the aggressive behavior: What does it means? It is unclear. I believe that also gynes and male could compete for carbohydrates as they use the same food.

….workers could be affected by outdoor activities.: I believe that the sentence should be: ….workers could be involved by outdoor activities

Behavioral description, intensity scores, winning percentages, and aggressive behavior trends

….radar chart of the number of each behavior. The score for each behavior is derived from Table 1 and Figure 2. The abbreviation of each behavior is as follows: In threatening, rushing opponent is TR, lifting antennae and front legs and shaking wings is TL, opening mandible is TO, threateningly flying over opponent is TT, and chasing opponent is TC. In Grappling, pushing or fighting while flying is GP, banging or pushing with head is GB, chasing and grabbing is GC, forcing down and throwing opponent is GF, and getting opponent and biting or stinging is GG (see Table 1).:

In my opinion, all this part is a little confusing. The table helps in distinguishing the various behaviors within the 3 categories win, lose, tie, however it is necessary to motivate the arbitrary assignment of the score based on the escalation of aggressive behavior. All the abbreviations are not so immediate and also they confuse the reading of the radar chart. Moreover, it does not seem to me that the abbreviations are an acronym for the category.

Morphological measurements

…We failed to collect all the individuals observed to determine their size, as aggressive individuals are often active and flew away immediately following the behavioral interactions. Therefore, body size was measured for 30 individuals per species.

This point may represent a methodological problem that could have biased the results. If, as the authors say, aggressive individuals were active and flew away immediately and only the losers were caught, it is likely that the size difference between the species is not representative of the real situation. It is possible that the most aggressive individuals were also the largest and the relationship between losers and winners within a species is not necessarily linear. Perhaps the authors should say how many of these 30 individuals by species were "winners and losers"

Results

Aggressive intensity, winning percentages, and aggressive behavior trends

This whole paragraph of the results in my opinion is heavy to read. For each of the 5 comparisons there is a succession of numbers, percentages and acronyms. I would suggest to add another table; however, the number of wins / ties / loses is also reported in figure 3

Discussion

…. V. velutina invaded in 2004 : it is more correct to say “was introduced”

….there was only native hornet species, V. crabro,… : In Europe there are two species of hornets: V. crabro and V. orientalis. The latter should be metioned too.

…..For example, stag beetle males have larger mandibles, and larger crickets have higher aggressiveness and higher RHP [34,35,43,44]. Larger fish nests and fish males have higher RHP and higher reproductive success rates [45], and larger invasive fish have higher RHP than small native fish [46]. Therefore, in this study, as V. velutina is a similar size to or larger than V. simillima, but smaller in size than the other four native hornets, its RHP seems to be relatively low, as shown in the aggressiveness results of this study. : This part needs to be rewritten better. It looks as a series of info thrown there and a little disconnected from each other.

Fig. 1 the image quality is rather poor. I suggest to improve it

Fig. 2 How is the average of the aggressiveness score calculated? I suppose it is the average of the scores in the table for each species, but I believe that it should be better specified.

fig 3 It should be improved in quality. The abbreviations "vel & sim" etc. should be changed; please put the exact percentages rather the rounded value (e.g. velutin against mandarinia: 3.9%, 4.8% and 91.3%)

Fig 4 The abbreviation KK appears in the radar chart but it is not mentioned among the other acronyms

6. PLOS authors have the option to publish the peer review history of their article (what does this mean?). If published, this will include your full peer review and any attached files.

Reviewer #1: Yes: Denis Thiery

Reviewer #2: No

---

## [Author Response · Author response to Decision Letter 0]

19 Mar 2020

Reviewer 1. 

Question 1: What about the invasive status of the other hornets.

Answer: In Korea, velutina is the first study of exotic wasps, so existing exotic wasps have no official record. Therefore, the following modifications were made.

In South Korea, the social and agricultural impacts are also gradually increasing due to invasions by black widow spiders (Latrodectus hesperus), spotted lanternfly (Lycorma delicatula), frosted moth-bug (Metcalfa pruinosa), and black planthopper (Ricania speculum). In particular, social insects such as yellow-legged hornets (Vespa velutina), red imported fire ants (Solenopsis invicta), and argentine ant (Linepithema humile), which are particularly damaging due to the large number of individuals and their toxicities, are recent introductions to Korea and pose a significant social threat. [6–8, 63-66]. 

63. Lee HS, im DE, Lyu DP. 2020. Discovery of the Invasive Argentine ant, Linepithema humile (Mayr)(Hymenoptera: Formicidae: Dolichoderinae) in Korea. Korean J. Appl. Entomol. 59(1) 71-72

64. Han JM, Kim H, Lim EJ, Lee S, Kwon YJ, Cho S. 2008. Lycorma delicatula (Hemiptera: Auchenorrhyncha: Fulgoridae: Aphaeninae) finally, but suddenly arrived in Korea. Entomological Research 38(4): 281-286

65. Kim Y, Kim M, Hong KJ, Lee S. 2011 Outbreak of an exotic flatid, Metcalfa pruinosa (Say) (Hemiptera: Flatidae), in the capital region of Korea. Journal of Asia-Pacific Entomology. 14: 473-478

66. Choi DS, Kim DI, Ko SJ, Kang BR, Lee KS, Park JD, Choi KJ. 2012. Occurrence ecology of Ricania sp. (Hemiptera: Ricaniidae) and selection of environmental friendly agricultural materials for control. Korean Journal of Applied Entomology. 51(2): 141-148.

Question 2: About the health impact. This impact on health is to my opinion not completely founded. A study in france (De haro et al.) published that cases of hymenopteran evenomations did not increase in france after Vv invasion, and only two cases of death were clearly identified (venom identified by the police scientific services). The main impact on health is of course due to shock in case of multiple sting when attacking a nest or allergic reaction (same with honey bees or yellow jackets).

What coul also be mentionned is the frenzy to citizens when nests are in private gardens or parks.

This § could be improved details concerning mandatory in Korea should not appear in the full text since it concerns only South Korea. Could this be in additional data ?

Answer: I have made the following modifications to reflect your opinion.

In addition, V. velutina is a poisonous insect that has a public health impact. In Korea, there are more than 100,000 cases of removal of social wasps’ nests per year, and V. velutina’s nest removal rate is the highest among Vespa species. The average number of injuries caused by social wasps is about 15,000, and there have been about 10 deaths. IIn particular, due to the high density in urban areas, the damage caused by them is likely to be high [14,15,24]. In fact, two deaths have occurred in France since the invasion of V. velutina, and two deaths have been reported in Korea. At present, the impacts of V. velutina may not be noticeable [69], but the actual impacts are expected to be more because the extent of the impacts that this alien species causes may not be fully appreciated [23, 67, 68, 70]. 

67. Park JJ, Jung C. 2016. Risk prediction of the distribution of invasive hornet, Vespa velutina nigrithorax in Korea using CLIMEX model. J Apic 31(4): 293-303.

68. Lioy S, Manino A, Porporato M, Laurino D, Romano A, Capello M, Bertolino S (2019) Establishing surveillance areas for tackling the invasion of Vespa velutina in outbreaks and over the border of its expanding range. NeoBiota 46: 51-69.

69. de Haro L, Labadie M, Chanseau P, Cabot C, Blanc-Brisset I, Penouil F, National Coordination Committee for Toxicovigilance. 2010. Medical consequences of the Asian black hornet (Vespa velutina) invasion in Southwestern France. Toxicon 55: 650-652.

70. Monceau K and Thiery D. 2017. Vespa velutina nest distribution at a local scale: An 8-year survey of the invasive honeybee predator. Insect. Sci. 24(4): 663-674.

Question 3: I do not fully agree with one of the last intro sentence : 'in particular, the possibility and speed of IAS spread... are detremined by the hierarchy...' Not only, biocontrol agents localy presents and natural ennemies may also wipe out the IAS during the first steps.This should be rephrased (please quote example of IAS control by natural ennemies (including disease). This values also for the begining of discussion.

Answer : I have made the following modifications to reflect your opinion.

Several factors determine the rate of spread of an invasive species. In addition to anthropogenic controls [1, 71, 73] and natural enemies [74-75], the ecological hierarchy obtained through competition among similar species makes an important contribution to the likelihood and speed of range extension. 

71. Simberloff D, Martin JL, Genovesi P, Maris V, Wardle DA, Aronson J, Courchamp F, Galil B,. Garcia-Berthou E., Pascal M., Pysek P, Sousa R. Tabacchi E., Vila M. 2013. Impacts of biological invasions: what’s what and the way forward. Trends in Eco & Evol. 28(1): 58-66.

73. Carrasco LR Baker R, MacLeod A, Knight JD, Mumford D. 2010. Optimal and robust control of invasive alien species spreading in homogeneous landscapes. J R Soc Interface. 7: 529-540.

74. Toepfer S. Kuhlmann U. 2004. Survey for natural enemies of the invasive alien chrysomelid, Diabrotica virgifera virgifera, in Central Europe. BioControl. 49: 385-395.

75. Villemant C., Zuccon D., Rome Q, Muller F, Poinar GO, Justine JL (2015), Can parasites halt the invader? Mermithid nematodes parasitizing the yellow-legged Asian hornet in France. PeerJ 3:e947.

Question 4: The role of venom gland (as so caleld alarm pheromone in two recent papers should also be documented and discussed somewhere.

Answer : I made the following corrections to the discussion part to fully reflect your opinion.

Altering the concentration and composition of alarm pheromones in areas it has invaded also creates an effective defense strategy against potential predators [76-77]. 

76. Thiéry, D. Bonnard, O., Riquier, L, de Revel, G., Monceau, K. 2018. An alarm pheromone in the venom gland of Vespa velutina: evidence revisited from the european invasive population. Entomol Gen. 38(2): 145-156.

77. Cheng, Y, Wen P, Dong S, Tan K. Nieh JC 2017. Poison and alarm: The asian hornet Vespa velutina uses sting venom volatiles as alarm pheromone. J Exp Biol. 220: 645-651.

Reviewer 2

Question 1: ABSTRACT: The abstract needs to be rewritten by summarizing in a few lines the results by cutting all numbers and percentage of fights.

Answer : I have made the following modifications to reflect your opinion.

The range of the invasive alien hornet, Vespa velutina nigrithorax, has been expanding since its introduction to Korea in 2003. Here, we compare the aggressive behaviors and body size of V. velutina nigrithorax with five native hornet species to identify the interspecific hierarchies that influence the rate of spread of this species. Aggressive behaviors were classified into 11 categories, and each interaction was scored as a win, loss, or tie. We found that V. velutina was superior to V. simillima in fights that V. velutina won and showed a high incidence of threatening behavior. V. mandarinia outperformed V. velutina in fights that V. mandarinia won and grappling behavior was common. V. analis was superior to V. velutina in fights that V. analis won and showed a high degree of threatening behavior. V. crabro was superior to V. velutina in fights that V. crabro won and showed a high rate of threatening behavior. V. dybowskii was superior to V. velutina in fights that V. dybowskii won and showed a high incidence of threatening and grappling behaviors. The body size of V. velutina was greater than V. simillima (although not statistically significant) and smaller than all other Vespa species. Therefore, according to this study, the low interspecific hierarchies of V. velutina seem to be a major cause of the slower spread in Korea than in Europe. However, over time, its density has gradually increased within the forest, where it seems to be overcoming its disadvantages and expanding its range, possibly because the large colonies and good flying abilities of this species help it secure food. 

Question 2: METHODS: Behavioral observation experiment

…..in a forest where hornets were active: what do you mean for “active”? I think it means “present”.”

Answer : I have made the following modifications to reflect your opinion.

The behavioral observation experiment apparatus was set up as follows: a table (1m in height) was installed on a flat plain in a forest where hornets were present, 2-3 sheets of toilet paper were placed on top of the table, and an attractant was poured onto these,....

Question 3: ….to spray the attractant for approximately 10-20 minutes : Please, give the quantity of the attractant sprayed. I believe that it should be better reported the composition here rather than the reference that reported it.

It is unclear if the Authors sprayed the attractant every 10-20 min or they spayed the attractant in the air for 10-20 min before beginning the observations to increase the hornets attraction?

Moreover, I have some concern about the attractant used. It seems to be a generic one but the different species could be differently specialized as reported by Matsura (1991). This could bias the experiment as one species could be more aggressive than others if it must defence a resource more . Have the Authors some data about the attractiveness level for each species?

Answer : I have made the following modifications to reflect your opinion.

In addition, to increase hornet attraction, a 500-ml nebulizer was filled with attractant, and the liquid was sprayed for approximately 10-20 minutes before observations began. The attractant was composed of 1: 1: 1 brown sugar water, vinegar, and ethanol, which mimics oak sap. It is the most commonly used substance for attracting and capturing Vespinae species in hornet traps in Korea, and it has very little attractiveness bias for a particular species [14, 72].

72. Chio MB, Park BA, Lee JW. 2012c. The Species Diversity and Distribution of Vespidae in Southeast Region (Sangdong-eup, Gimsatgat-myeon, Jungdong-myeon) of Yeongwol-gun, Gangwon-do, Korea. J Korean Nat. 5(4): 305-310.

Question 4: ….. because new queens or males were confused with the aggressive behavior: What does it means? It is unclear. I believe that also gynes and male could compete for carbohydrates as they use the same food.

Answer : I have made the following modifications to reflect your opinion.

This experiment was conducted only with workers, excluding gynes and males, to get rid of the differences in attack behaviors due to caste differences. Therefore, we conducted our study over a total of four days from August 12-13 and August 17-18, 2017, before the gynes and males came out in mid-September. 

Question 5: ….workers could be affected by outdoor activities.: I believe that the sentence should be: ….workers could be involved by outdoor activities

Answer : I have made the following modifications to reflect your opinion.

The experiment was conducted at 8-10 am and 5-7 pm because workers tend to avoid outdoor activities when daytime temperatures exceed 35 degrees. 

Question 6: Behavioral description, intensity scores, winning percentages, and aggressive behavior trends

….radar chart of the number of each behavior. The score for each behavior is derived from Table 1 and Figure 2. The abbreviation of each behavior is as follows: In threatening, rushing opponent is TR, lifting antennae and front legs and shaking wings is TL, opening mandible is TO, threateningly flying over opponent is TT, and chasing opponent is TC. In Grappling, pushing or fighting while flying is GP, banging or pushing with head is GB, chasing and grabbing is GC, forcing down and throwing opponent is GF, and getting opponent and biting or stinging is GG (see Table 1).:

In my opinion, all this part is a little confusing. The table helps in distinguishing the various behaviors within the 3 categories win, lose, tie, however it is necessary to motivate the arbitrary assignment of the score based on the escalation of aggressive behavior. All the abbreviations are not so immediate and also they confuse the reading of the radar chart. Moreover, it does not seem to me that the abbreviations are an acronym for the category.

Answer: Based on your opinion, I have modified Figure 4 to make it easier to see.

Question 7: Morphological measurements

…We failed to collect all the individuals observed to determine their size, as aggressive individuals are often active and flew away immediately following the behavioral interactions. Therefore, body size was measured for 30 individuals per species.

This point may represent a methodological problem that could have biased the results. If, as the authors say, aggressive individuals were active and flew away immediately and only the losers were caught, it is likely that the size difference between the species is not representative of the real situation. It is possible that the most aggressive individuals were also the largest and the relationship between losers and winners within a species is not necessarily linear. Perhaps the authors should say how many of these 30 individuals by species were "winners and losers“

Answer: This means that during the 67-153 fight between each of the two species, most of the individuals were captured but some missed cases. In that case the remaining subjects were not included in the size measurement.

And because the number of fights between each species is very high and more than 90% of cases have collected both winers and losers, they do not significantly affect the results. In addition, the sample of 30 representative battles was selected and measured, which does not affect the results.

However, as you said, the contents of the text could lead to such misunderstandings, so this sentence was deleted and simply rewritten as follows.

Both winners and losers were included in the sampling. If the test subjects were missed, the remaining opponents were excluded from the measurement.

Body size of the five native species was measured by selecting 30 individuals from the samples collected during the experiment. In addition, V. velutina was selected from the individuals who fought the five native species, and a total of 30 individuals were measured. 

Question 8: Results

Aggressive intensity, winning percentages, and aggressive behavior trends

This whole paragraph of the results in my opinion is heavy to read. For each of the 5 comparisons there is a succession of numbers, percentages and acronyms. I would suggest to add another table; however, the number of wins / ties / loses is also reported in figure 3

Answer: In response to your opinion, the hard to read parts were deleted and easy to read. And the deleted content has already been explained in Fig. 2, so no additional table was created.

Aggressive intensity, winning percentages, and aggressive behavior trends

In V. velutina vs. V. simillima, V. velutina was superior to V. simillima in 153 fights (P < 0.001, Fig. 2A), and won 71% of the encounters (Fig . 3). V. velutina had a high rate of threatening behavior, such as rushing the opponent, and lifting the antennae and front legs and shaking the wings. V. simillima, however, rushed the opponent more often than V. velutina (Fig. 4A). 

In V. velutina vs. V. mandarinia, V. mandarinia outperformed V. velutina in a total of 104 fights (P < 0.001, Fig. 2B) and won 91% of the encounters (Fig. 3). V. mandarinia displayed a high rate of grappling behavior, such as banging or pushing with the head (Fig. 4B). V. mandarinia also preyed upon V. velutina, although rarely.

In V. velutina vs. V. analis, V. analis was superior to V. velutina in 67 fights (P < 0.001, Fig. 2C) and won 76% of the encounters (Fig. 3). Concerning threatening behavior, V. analis displayed threat most often by rushing the opponent, and this species’ second favorite ploy was to fly threateningly over the adversary. In contrast, V. velutina made a weak show of rushing the opponent (Fig. 4C). 

In contests between V. velutina and V. crabro, V. crabro was superior to V. velutina in 93 fights (P < 0.001, Fig. 2D) and won 73% of the encounters (Fig. 3). Lifting antennae and front legs and shaking wings were the most common threatening behaviors in V. crabro, and banging or pushing with the head were the most common grappling behaviors. In contrast, V. velutina displayed minor aggressive behavior such as threateningly flying over the opponent (Fig. 4D).

Finally, in encounters between V. velutina and V. dybowskii, V. dybowskii was superior to V. velutina in a total of 132 fights (P < 0.001, Fig. 2E) and won 91% of the contests (Fig. 3). The most common threatening behavior employed by V. dybowskii was chasing the opponent, and the second most common action was rushing the opponent; chasing and grabbing were the most common grappling behaviors for this species. V. velutina’s counterattack action was to fly threateningly over the opponent but not vigorously or often (Fig. 4E). 

Question 9: Discussion

…. V. velutina invaded in 2004 : it is more correct to say “was introduced”

….there was only native hornet species, V. crabro,… : In Europe there are two species of hornets: V. crabro and V. orientalis. The latter should be metioned too.

Answer: Based on your opinion, I made the following modifications.

In Europe, where V. velutina was introduced in 2004, there were already two native hornet species, V. crabro and V. orientalis, 

Thus, in Europe where the competition was V. crabro at the beginning of the invasion [25,26], 

Question 10: …..For example, stag beetle males have larger mandibles, and larger crickets have higher aggressiveness and higher RHP [34,35,43,44]. Larger fish nests and fish males have higher RHP and higher reproductive success rates [45], and larger invasive fish have higher RHP than small native fish [46]. Therefore, in this study, as V. velutina is a similar size to or larger than V. simillima, but smaller in size than the other four native hornets, its RHP seems to be relatively low, as shown in the aggressiveness results of this study. : This part needs to be rewritten better. It looks as a series of info thrown there and a little disconnected from each other.

Answer: For example, in various insects, larger individuals in the intraspecific have higher aggressiveness and higher RHP than small ones, indicating that they are competitive [34, 35, 43, 44]. Large invasive species have been shown to spread more easily in nature with competitive advantages at higher RHP than smaller native species [46]. 

Question 11: Fig. 1 the image quality is rather poor. I suggest to improve it

Answer: Due to the nature of the behavioral experiments, except for a few photos, most of the videos were captured and used, so there was a little shaky picture, but there would be no big problem to explain. The pixels of the image to be submitted are high.

Question 12: Fig. 2 How is the average of the aggressiveness score calculated? I suppose it is the average of the scores in the table for each species, but I believe that it should be better specified.

Answer: The data was provided in the supplementary table.

S1 table. Distribution of aggressiveness scores between an invasive alien hornet, Vespa velutina, and five native Korean hornet species

Score vel & sim vel & man vel & ana vel & cra vel & dyb

2 : 0 73 3 10 13 7

3 : 0 31 1 2 3 1

4 : 0 4 0 0 0 0

1 : 1 10 5 4 9 4

0 : 2 28 39 39 49 73

0 : 3 7 42 12 19 42

0 : 4 0 11 0 0 5

0 : 5 0 3 0 0 0

total 153 104 67 93 132

Question 13: fig 3 It should be improved in quality. The abbreviations "vel & sim" etc. should be changed; please put the exact percentages rather the rounded value (e.g. velutin against mandarinia: 3.9%, 4.8% and 91.3%)

Answer: I modified it as you suggested.

Question 14: Fig 4 The abbreviation KK appears in the radar chart but it is not mentioned among the other acronyms

Answer: As I said before, I modified the figure 4 to make it easier to see.

---

## [Decision Letter · Decision Letter 1]

7 May 2020

PONE-D-19-34041R1

Interspecific hierarchies from aggressiveness and body size among the invasive alien hornet, Vespa velutina nigrithorax, and five native hornets in South Korea

PLOS ONE

Dear Dr. Choi,

Thank you for submitting your manuscript to PLOS ONE. After careful consideration, we feel that it has merit but does not fully meet PLOS ONE’s publication criteria as it currently stands. Therefore, we invite you to submit a revised version of the manuscript that addresses the points raised during the review process.

ACADEMIC EDITOR: The manuscript has been revised by one of the previous reviewers that consider that the authors have done a good job including all the suggestions and the manuscript is acceptable now for publication. However, I have some minor comments that should be addressed before final acceptance:

-This is a minor, but VERY IMPORTANT comment: All the P-values reported need to be associated to their statistical (e.g. t, F) and degrees of freedom (df). To report properly the statistical analyses all these features should be given in the text (or in tables in main text) together with the P value. Examples:  t(28) = 2.6, p < .05 or F_2,28_= 55, P=0.01. These values should correspond to the t-tests or ANOVA global test, while the Tuckey a posteriori test should be given in the figures using asterisks or letters (I see this has been done in one of the figures), to show significant differences between different levels of a significant factor. In case of body size you report the F but not the df. Also this sentence should be slightly rewritten because the result of anova might give whether species (in general) differ in size, whereas it is the post hoc test that show which is different from which (which is not totally clear in the way it is written).

-Regarding the figures, it is not necessary to write in each panel ‘mean ±SD’. Instead, say it in the legend, and show their significance (using for instance asterisk that are mentioned in the legend).

-Check also for small typological mistakes that appear along the text.

We would appreciate receiving your revised manuscript by Jun 21 2020 11:59PM. To enhance the reproducibility of your results, we recommend that if applicable you deposit your laboratory protocols in protocols.io, where a protocol can be assigned its own identifier (DOI) such that it can be cited independently in the future. For instructions see: http://journals.plos.org/plosone/s/submission-guidelines#loc-laboratory-protocols

We look forward to receiving your revised manuscript.

Kind regards,

Amparo Lázaro, PhD

Academic Editor

PLOS ONE

Reviewers' comments:

Reviewer's Responses to Questions

**Comments to the Author**

1. If the authors have adequately addressed your comments raised in a previous round of review and you feel that this manuscript is now acceptable for publication, you may indicate that here to bypass the “Comments to the Author” section, enter your conflict of interest statement in the “Confidential to Editor” section, and submit your "Accept" recommendation.

Reviewer #2: All comments have been addressed

2. Is the manuscript technically sound, and do the data support the conclusions?

Reviewer #2: (No Response)

3. Has the statistical analysis been performed appropriately and rigorously? 

Reviewer #2: (No Response)

4. Have the authors made all data underlying the findings in their manuscript fully available?

Reviewer #2: (No Response)

5. Is the manuscript presented in an intelligible fashion and written in standard English?

Reviewer #2: (No Response)

6. Review Comments to the Author

Reviewer #2: (No Response)

7. PLOS authors have the option to publish the peer review history of their article (what does this mean?). If published, this will include your full peer review and any attached files.

Reviewer #2: No

---

## [Author Response · Author response to Decision Letter 1]

16 Jun 2020

Question 1: All the P-values reported need to be associated to their statistical (e.g. t, F) and degrees of freedom (df). To report properly the statistical analyses all these features should be given in the text (or in tables in main text) together with the P value. Examples: t(28) = 2.6, p < .05 or F2,28= 55, P=0.01. These values should correspond to the t-tests or ANOVA global test, while the Tuckey a posteriori test should be given in the figures using asterisks or letters (I see this has been done in one of the figures), to show significant differences between different levels of a significant factor. In case of body size you report the F but not the df. Also this sentence should be slightly rewritten because the result of anova might give whether species (in general) differ in size, whereas it is the post hoc test that show which is different from which (which is not totally clear in the way it is written).

Response: All the degrees of freedom mentioned by the reviewer were inserted.

The revised part is indicated in red letters.

Aggressive intensity, winning percentages, and aggressive behavior trends

In V. velutina vs. V. simillima, V. velutina was superior to V. simillima in 153 fights (t(304)=9.89, P < 0.001, Fig. 2A), and won 71% of the encounters (Fig. 3). V. velutina had a high rate of threatening behavior, such as rushing the opponent, and lifting the antennae and front legs and shaking the wings. V. simillima, however, rushed the opponent more often than V. velutina (Fig. 4A).

In V. velutina vs. V. mandarinia, V. mandarinia outperformed V. velutina in a total of 104 fights (t(206)=-22.75, P < 0.001, Fig. 2B) and won 91% of the encounters (Fig. 3). V. mandarinia displayed a high rate of grappling behavior, such as banging or pushing with the head (Fig. 4B). V. mandarinia also preyed upon V. velutina, although rarely.

In V. velutina vs. V. analis, V. analis was superior to V. velutina in 67 fights (t(132)=-8.38, P < 0.001, Fig. 2C) and won 76% of the encounters (Fig. 3). Concerning threatening behavior, V. analis displayed threat most often by rushing the opponent, and this species’ second favorite ploy was to fly threateningly over the adversary. In contrast, V. velutina made a weak show of rushing the opponent (Fig. 4C).

In contests between V. velutina and V. crabro, V. crabro was superior to V. velutina in 93 fights (t(184)=-9.62, P < 0.001, Fig. 2D) and won 73% of the encounters (Fig. 3). Lifting antennae and front legs and shaking wings were the most common threatening behaviors in V. crabro, and banging or pushing with the head were the most common grappling behaviors. In contrast, V. velutina displayed minor aggressive behavior such as threateningly flying over the opponent (Fig. 4D).

Finally, in encounters between V. velutina and V. dybowskii, V. dybowskii was superior to V. velutina in a total of 132 fights (t(262)=-24.22, P < 0.001, Fig. 2E) and won 91% of the contests (Fig. 3). The most common threatening behavior employed by V. dybowskii was chasing the opponent, and the second most common action was rushing the opponent; chasing and grabbing were the most common grappling behaviors for this species. V. velutina’s counterattack action was to fly threateningly over the opponent but not vigorously or often (Fig. 4E).

Fig. 2. Aggressiveness scores between an invasive alien hornet, V. velutina, and five native Korean hornet species. Scores are displayed as mean values ± SD. A: V. velutina (score 1.73±1.11) vs Vespa simillima (score 0.57±0.94), t(304)=9.89, P<0.001; B: V. velutina (score 0.13±0.48) vs Vespa mandarinia (score 2.58±0.98), t(206)=-22.75, P<0.001; C: V. velutina (score 0.45±0.86) vs Vespa analis (score 1.76±0.96), t(132)=-8.38, P<0.001; D: V. velutina (score 0.47±0.86) vs Vespa crabro (score 1.76±0.97), t(184)=-9.62, P<0.001; E: V. velutina (score 0.16±0.54) vs Vespa dybowskii (score 2.24±0.83), t(262)=-24.22, P<0.001. (See S1 table).

Body size

The body size of V. velutina was 8.04 ± 0.41 mm, which was slightly larger than V. simillima 7.80 ± 0.29 mm. V. dybowskii was 8.88 ± 0.49 mm, V. analis was 10.14 ± 0.51 mm, V. crabro was 9.82 ± 0.4 mm, and V. mandarinia was 13.21 ± 0.83 mm. Thus, except for V. simillima, all species were larger than V. velutina. These differences were all significant (F(5,174)=434.9, P<0.001),except for those between V. simillima and V. velutina, and V. analis and V. crabro (Fig. 5).

Fig. 5. Body size differences between an invasive alien hornet, V. velutina and five native Korean hornet species. (F(5,174)=434.9, P<0.001).

Question 2: Regarding the figures, it is not necessary to write in each panel ‘mean ±SD’. Instead, say it in the legend, and show their significance (using for instance asterisk that are mentioned in the legend).

Response: Delete ‘mean ±SD’ in the figure and indicate significance in alphabet

Question 3: Check also for small typological mistakes that appear along the text.

Response: Some modifications were made to the italics and notation of references.

---

## [Editor Report · Decision Letter 2]

19 Jun 2020

PONE-D-19-34041R2

Interspecific hierarchies from aggressiveness and body size among the invasive alien hornet, Vespa velutina nigrithorax, and five native hornets in South Korea

PLOS ONE

Dear Dr. Choi,

Thank you for submitting your manuscript to PLOS ONE. After careful consideration, we feel that it has merit but does not fully meet PLOS ONE’s publication criteria as it currently stands. Therefore, we invite you to submit a revised version of the manuscript that addresses the points raised during the review process.

Overall, the authors have addressed adequately the minor changes proposed, but I just have a small suggestion to write more adequately  the results of body size before final acceptance:

The text of body size is a bit confusing mostly because of this sentence: ‘These differences were all significant (F(5, 174) = 434.9, P <0.001), while their post hoc test showed no statistic differences between V. simillima and V. velutina, and V. analis and V. crabro (Fig. 5)’, which is unclear because you say that ALL are different and then that some are not. The first test shows whether there are differences among species in size (not that all the species differ between them), and the posthocs which species do differ between them.

I suggest writing this as follows (or in a similar manner):

 ‘There were significant differences in body size among Vespa species (F(5, 174) = 434.9, P <0.001), being the differences significant between all the species except for V. simillima and V. velutina, and V. analis and V. crabro (Fig. 5)’

We look forward to receiving your revised manuscript.

Kind regards,

Amparo Lázaro, PhD

Academic Editor

PLOS ONE

---

## [Author Response · Author response to Decision Letter 2]

22 Jun 2020

Modified "There were significant differences in body size among Vespa species (F(5, 174) = 434.9, P <0.001), being the differences significant between all the species except for V. simillima and V. velutina, and V. analis and V. crabro (Fig. 5)".

---

## [Editor Report · Decision Letter 3]

25 Jun 2020

Interspecific hierarchies from aggressiveness and body size among the invasive alien hornet, Vespa velutina nigrithorax, and five native hornets in South Korea

PONE-D-19-34041R3

Dear Dr. Choi,

We’re pleased to inform you that your manuscript has been judged scientifically suitable for publication and will be formally accepted for publication once it meets all outstanding technical requirements.

Kind regards,

Amparo Lázaro, PhD

Academic Editor

PLOS ONE
---

## [Editor Report · Acceptance letter]

6 Jul 2020

PONE-D-19-34041R3 

Interspecific hierarchies from aggressiveness and body size among the invasive alien hornet, Vespa velutina nigrithorax, and five native hornets in South Korea 

Dear Dr. Choi:

I'm pleased to inform you that your manuscript has been deemed suitable for publication in PLOS ONE. Congratulations! Your manuscript is now with our production department. 

Kind regards, 

on behalf of

Dr. Amparo Lázaro 

Academic Editor

PLOS ONE